# Thromboembolic Events in Patients with Influenza: A Scoping Review

**DOI:** 10.3390/v14122817

**Published:** 2022-12-17

**Authors:** Raffaella Rubino, Claudia Imburgia, Silvia Bonura, Marcello Trizzino, Chiara Iaria, Antonio Cascio

**Affiliations:** 1Infectious and Tropical Diseases Unit, AOU Policlinico “P-Giaccone”, 90127 Palermo, Italy; 2Infectious Diseases Unit, ARNAS Civico, 90127 Palermo, Italy; 3Department of Health Promotion, Maternal and Infant Care, Internal Medicine and Medical Specialties, University of Palermo, 90133 Palermo, Italy

**Keywords:** thrombosis, stroke, influenza, flu, thromboembolism, infarct

## Abstract

Introduction: Influenza is an acute respiratory infection that usually causes a short-term and self-limiting illness. However, in high-risk populations, this can lead to several complications, with an increase in mortality. Aside from the well-known extrapulmonary complications, several studies have investigated the relationship between influenza and acute cardio and cerebrovascular events. Reviews of the thromboembolic complications associated with influenza are lacking. Objectives: the study aims to conduct a scoping review to analyze the epidemiological and clinical characteristics of patients suffering from influenza and thromboembolic complications. Materials and methods: A computerized search of historical published cases using PubMed and the terms “influenza” or “flu” and “thrombosis”, “embolism”, “thromboembolism”, “stroke”, or “infarct” for the last twenty-five years was conducted. Only articles reporting detailed data on patients with thromboembolic complications of laboratory-confirmed influenza were considered eligible for inclusion in the scoping review. Results: Fifty-eight cases with laboratory documented influenza A or B and a related intravascular thrombosis were retrieved. Their characteristics were analyzed along with those of a patient who motivated our search. The localizations of thromboembolic events were pulmonary embolism 21/58 (36.2%), DVT 12/58 (20.6%), DVT and pulmonary embolism 3/58 (5.1%), acute ischemic stroke 11/58 (18.9%), arterial thrombosis 4/58 (6.8%), and acute myocardial infarction 5/58 (8.6%). Discussion: Our findings are important in clarifying which thromboembolic complications are more frequent in adults and children with influenza. Symptoms of pulmonary embolism and influenza can be very similar, so a careful clinical evaluation is required for proper patient management, possible instrumental deepening, and appropriate pharmacological interventions, especially for patients with respiratory failure.

## 1. Introduction

Influenza is an acute respiratory infection caused by influenza viruses (A or B) which are spread from person to person primarily through droplets and during the winter season. Annually, influenza infects approximately 10% to 20% of the world’s population causing mostly a non-complicated, self-limiting disease [1]. Influenza outbreaks occur in two distinct patterns: endemic (seasonal) and pandemic. Pandemics are typically decades apart and are linked to the influenza A virus strain with novel hemagglutinin molecule forms [2,3].

The virulence of influenza A viruses depends on antigenic drift or reassortment between human and animal influenza A viruses that allows the virus to evade host immunity [4]. In high-risk populations, influenza can cause several complications, with an increase in mortality. Obesity, pregnancy, immunosuppression, chronic pulmonary diseases, and low socioeconomic status, which limit access to medical care, are major risk factors for increased influenza-associated morbidity and mortality among infected individuals [5]. The main complications are localized in the respiratory tract, such as primary influenza pneumonia or secondary bacterial pneumonia, and can cause acute respiratory distress syndrome (ARDS), but the true burden of influenza infection is likely much higher because, in addition to well-described extrapulmonary complications, several studies have examined the relationship between influenza and acute cardio and cerebrovascular events and found convincing evidence [1,6]. 

There is also mounting evidence that acute infections are linked to an increased risk of developing venous thromboembolic events [7,8]. Deep-vein thrombosis and pulmonary embolism have an annual incidence of 2–3 per 1000 people, and many risk factors, such as genetic predisposition, immobilization, surgery, pregnancy, oral contraceptives, and malignancies have been identified. The presence of both venous and arterial thrombosis during influenza raises the possibility that the mechanisms causing thrombus formation are distinct from those causing venous thrombosis alone [9]. The pathogenesis of arterial thrombosis differs from that of venous thrombosis, and it typically occurs as a result of endovascular injury or endothelial activation caused by the elaboration of a variety of proinflammatory mediators with subsequent platelet and coagulation pathway activation. Even though extrapulmonary complications associated with influenza have been well-documented over the years, an updated review of the thromboembolic complications is lacking. 

The aim of this paper is to analyze the epidemiological and clinical characteristics of all patients with thromboembolic complications of influenza identified through a comprehensive literature search. 

A case we recently observed, which motivated our search, is also described, and its characteristics were analyzed together with those of all the other cases found in the systematic search.

CASE REPORT: In January 2018, a 38-year-old immunocompetent man was admitted to the Unit of Infectious Disease at the University Hospital of Palermo with fever, dyspnea, hemoptysis, and the occurrence of a fatigue episode. There was no mention of cancer, venous thromboembolism (VTE), other comorbidities, or risk factors for pulmonary embolism. He weighed 75 kg, and he had never received a seasonal influenza vaccination. At the time of admission, the patient was awake with a body temperature of 36 °C, a heart rate of 80 beats per minute, a respiratory rate of 36 breaths per minute, and a blood pressure of 140/90 mmHg. On room air, O_2_ saturation was 86%, and chest auscultation revealed a diffusely reduced murmur with bilateral crackles at the bases. The pH was 7.45. The pO_2_ was 51.8 mmHg, and the pCO_2_ was 40.2 mmHg, according to blood gas analysis. Laboratory tests showed leukopenia with white blood cells 2.700/mmc (neutrophils 65%; lymphocytes 27.6%), Hb 14.9 g/dL, platelets 197.000/mmc, C-reactive protein 30 mg/L (normal value < 5), serum procalcitonin 0.14 ng/dL, aspartate aminotransferase/alanine aminotransferase 127/75 U/L, creatine phosphokinase 1473 U/L, creatinine 0.94 mg/dL, D-dimer 415 µcg/L (normal value 10–250), and blood coagulation parameters within the normal range. A thrombophilia screening was not done.

On real-time reverse transcription-polymerase chain reaction, a nasopharyngeal swab for respiratory viruses tested positive for influenza A/H1N1, whereas sputum and blood cultures, as well as urinary antigen tests for Streptococcus pneumonia and Legionella pneumophila, were negative. Serology tests for HIV, Chlamydia pneumoniae, and Mycoplasma pneumoniae all came back negative. An ECG revealed sinus rhythm.

The patient was placed on a Venturi mask with an FiO_2_ of 31%, and a chest CT scan was obtained, which revealed bilateral and diffuse areas of lung parenchymal consolidation. 

The patient was started on oseltamivir 75 mg twice daily, antibiotics, and subcutaneous prophylactic enoxaparin 4000 UI once daily.

A D-dimer test was repeated, due to persistent hypoxia despite antiviral treatment, and significantly higher values were found (3604 µcg/L seven days after admission). Therefore, a computed tomography pulmonary artery examination was performed on the seventh day of hospitalization, revealing opacification defects in the left main branch, segmental branches of both the left upper and lower lobes, and the tributary branches of the posterior segment of the right lower lobe. On echocardiography, the right ventricle appeared normal.

In light of the above finding, fondaparinux 7.5 mg once daily was immediately administered. The patient improved and no longer required oxygen and was switched to an oral regimen containing dabigatran 150 mg twice daily. 

A computed tomography pulmonary artery examination was repeated a week later, revealing nearly complete resolution of both the embolic and parenchymal images. The patient was discharged with no complications. 

## 2. Literature Review

A computerized search of historical published cases was conducted using PubMed.

The search was conducted for the period including the last twenty-five years, using the following string terms: “influenza” [Title/Abstract] OR “flu” [Title/Abstract] AND “thrombosis” [Title/Abstract] OR “embolism” [Title/Abstract] OR “thromboembolism” [Title/Abstract] OR “stroke” [Title/Abstract] OR “infarct” [Title/Abstract]. Furthermore, all references were hand-searched for additional relevant articles, and a citation tracker was used to identify any additional relevant literature. 

The most recent search was executed in May 2022, and the search results were case reports and case series.

Articles published in languages other than English, French, or Italian were not considered. The selected articles were separately reviewed by two independent authors (A.C. and C.I.). 

If an article reported detailed data on patients with thromboembolic complications of a laboratory-documented influenza, it was considered eligible for inclusion in the systematic review. We also included children and people under the age of 18.

For each patient included in the study, the following parameters were taken into account: sex, age, site of thrombosis, risk factors and/or co-morbidities, other complications different from thromboembolism, and outcomes.

The guidelines for Preferred Reporting Items for Systematic Review and Meta-Analysis (Moher et al. PRISMA) [10] and the PRISMA Extension for Scoping Reviews (PRISMA–Scr): Checklist and Explanation [11] were followed.

Figure 1 depicts a flow chart summarizing the literature research approach.

## 3. Results

Forty-one articles reporting 8086 cases of respiratory infection-related intravascular thrombosis (arterial and venous) were found. Eleven articles with a total of 8029 cases were excluded. One was excluded because there was no description of the characteristics of the population studied, and the cause of the stroke was determined to be secondary to hypotension [12]; the other 10 articles looked at patients who had a stroke or a myocardial infarction but whose characteristics were not described, and the respiratory tract infections included viruses other than influenza viruses [13,14,15,16,17,18,19,20,21,22]. Finally, 30 full-text articles containing 57 cases of patients with laboratory-documented influenza A or B and a related intravascular thrombosis (arterial and venous) diagnosed through instrumental evaluations were considered [9,23,24,25,26,27,28,29,30,31,32,33,34,35,36,37,38,39,40,41,42,43,44,45,46,47,48,49,50,51].

Table 1 analytically shows the epidemiological and clinical characteristics, complications, and outcomes of the study population.

Eight articles reported cases of influenza-related intravascular thrombosis in children (mean age 6.2 years), including a 16-year-old boy [43,44,45,46,47,48,49,50,51]. Thirty-six (62%) were males, and twenty-two (37.9%) were females. There were seven females and one male among the children under the age of 14. The median age was 38 years old and was calculated from fifty-seven patients because the age was not reported in one case. Adults had an average age of 42.2 years. Documented influenza A infections were found in 55/58 (94.8%) of cases. Positive PCR biomolecular test results for influenza A were found in 50/58 of the patients, while the diagnostic method was not reported in 5/58 of the patients [27,38,42,45,48]. Influenza B was found in 3/58 of the patients. In one case, the method was not reported, and in the other two, a positive antigen on a nasal swab was identified [36,38,41]. Except for eight cases, all influenza A cases were H1N1. Two cases were serologically identified as influenza A H3N2, while the remaining six were not [24].

An immunocompromising condition was reported in 5/58 (8.6%) of the patients [26,33,51].

The causes of immunosuppression varied. One patient had sarcoidosis and was receiving corticosteroids [26]. Another had cerebral palsy [51], and three patients had a hematological malignancy: One had lymphoma in CHOP. One had non-Hodgkin lymphoma, and another patient had acute myeloid leukemia [26,33]. Three patients had type II diabetes (5.1%) [32,35,41], and ten suffered from hypertension (17.2%) [24,26,28,34,40,41]. Nine patients were obese [25,36]; three had asthma/COPD [25,26], and three had dyslipidemia [33,34,41]. Chronic kidney disease affected only one patient [26]. A young twenty-eight-year-old patient was suffering from chronic heart failure and had a pacemaker [35], and 3/58 of the patients had chronic ischemic heart disease [32,34,40]. Coagulation disorders were reported in a small number of cases. In 1/58, a protein S deficiency (heterozygous K196E mutation) was identified [45], and in 2/58 (3.4%), a protein S low activity (45% and 50%-VN 57–125%) was discovered [35,48]. In 20/58 (34.4%), coagulation disorders were not identified, and in 35/58 (60.3%) of the patients, they were not reported. There was no evidence of a platelet disorder in any of the cases. The following autoantibodies were found: 1/58 with positive anticardiolipin IgG (25/5 U/mL) [33]; 1/58 with positive HLA A29 [48]; one patient developed an antiphospholipid syndrome, ANA titer 640 (normal 64), rheumatoid factor titer 2048 (normal 64, both IgM and IgA), and anti-SSA/Ro52, anti-SSA/Ro60, and anti-SSA/La. The titer of anticardiolipin autoantibodies rose fifteen days after admission to the hospital. Both IgG and IgM anticardiolipin were negative two years later, but the patient had symptoms consistent with pulmonary embolism long before she developed aCL. Concentrations of aCL also remained elevated for several months after discharge without causing symptoms [23]. In all remaining cases (54/58 of the patients—93 percent), screening was either not performed or resulted in a negative result.

In terms of modifiable risk factors, 8/50 of the adults (16%) were active smokers [26,32,34,39,40]; 1/58 were immobilized due to cerebral palsy [51]; 9/50 of cases (18%) were obese (BMI > 30) [25,26]; and 3/21 of the women (14.2%) were pregnant [33]. None of the women used contraception. None of the patients had a history of venous or arterial thrombosis.

The eight children were not immunocompromised. Five of them tested positive for H1N1 influenza A and three for non-typed influenza A. One of them had a basilar artery infarction that progressed to a locked-in syndrome [44]; another child with a protein S deficiency, caused by a heterozygous K196E mutation, suffered from a complete thrombosis of the superior mesenteric vein, necessitating a resection of the ischemic small intestine [45]. A patient with a low protein S activity (45%—normal value 57–125%) and positivity for HLA A29 presented with bilateral ischemic maculopathy with encephalitis progression [48]. Breker DA et al. [49] reported retinal and geniculate infarction, causing vision loss in a previously healthy thirteen-year-old girl, whereas Calzedda R et al. [46] described a case of a two-year-old Caucasian female admitted to the emergency department with an acute onset of fever, dry cough, left hemiparesis, speech difficulties, and persistent weakness associated with an ischemic stroke on MRI that was confirmed by magnetic resonance angiography, as well as the presence of H1N1 RNA copies (>500 copies/mL) and elevated IL 1-β and IL 6 levels in CSF samples. Bell et al. and Honorat R et al. both described an onset of fever and seizures, and a magnetic resonance angiography confirmed left middle cerebral artery infarction [43,47]. Finally, Javedani PP et al. described an acute infarction involving the right frontal, parietal, temporal, and occipital lobes, as well as hyper densities suggestive of thrombosed cortical veins in a four-month-old female who presented with fever, decreased oral intake, and limp appearance after antibiotic administration [50].

A 16-year-old boy was immunocompromised due to cerebral palsy; he tested positive for H1N1 and developed a deep venous thrombosis in his left leg [50]. Six of the young patients received antiviral therapy [44,45,46,47,48,49], with data missing in only two cases [43,50]. All pediatric patients survived, albeit with complications. Only one patient recovered completely [47], and in only one case, the data was not reported [50].

Several symptoms necessitated hospitalization. There were 18/58 (31.3%) of the patients who had flu-like symptoms but no respiratory failure [28,31,34,35,37,38,39,43,44,46,47,48,49], 31/58 of the patients (53.4%) with respiratory failure [9,24,25,26,27,29,30,32,33,34,40,51], 15/58 (25.8%) with neurological symptoms [25,28,30,31,35,37,38,42,43,44,46,47,48,49,50] our case, and 5/58 (8.6%) with gastrointestinal symptoms [36,39,41,45,48]. Two patients with neurological onset reported vision loss—one due to vaso-occlusive vasculitis [48] and the other to a combined retinal and lateral geniculate nucleus infarction [49]; the onset was not described in 18/58 (31.3%) of the remaining cases. A complicating bacterial pneumonia was diagnosed in 4/58 of the patients (6.9%) [29,33,35,44]. A radiological picture of ARDS was found in 14 of the 58 (24.1%) patients [25,26,28,30], whereas in 28/58 (48.2%) of the cases, chest X-rays or chest CT scans were not reported or not performed. Most patients had a radiological picture consistent with viral pneumonia; however, in four cases, the chest CT revealed reticulonodular infiltrates or extensive bilateral densities in the absence of a positive sputum culture [23,35,51] our case. On chest computed tomography, one patient had bilateral consolidation with thickened interlobular septa and bronchovascular bundles, cardiomegaly, and right pleural effusion, most likely due to heart failure [32].

Other complications occurred in 30/58 (51.7%) of the cases.

A total of 23/58 (39.6%) of the patients were admitted to intensive care units and underwent mechanical ventilation, and 5/58 (8.6%) underwent non-invasive mechanical ventilation.

Finally, 16 out of 58 (27.5%) patients died, and for some of them, the cause of death was not related to a thromboembolic event. In only one case, the outcome was not reported [37].

Table 2 shows comorbidities and sites of thromboembolic events in the study population: pulmonary embolism in 21/58 (36.2%), DVT in 12/58 (20.6%), DVT and pulmonary embolism in 3/58 (5.1%), acute ischemic stroke in 11/58 (18.9%), arterial thrombosis in 4/58 (6.8%), and acute myocardial infarction in 5/58 (8.6%). One patient had arterial thrombosis, as well as deep venous thrombosis and pulmonary embolism [40], and another had pulmonary embolism with cerebral deep venous thrombosis [38].

## 4. Discussion

Our patient had pulmonary embolism while on thromboprophylaxis for severe influenza-related pneumonia, which prompted the literature review. In 33/58 (56.8%) of the patients, no significant comorbidity or modifiable risk factors for thromboembolic events were reported. Except for a certain prevalence of the male sex (M/F 2.3:1) and of obese and hypertensive patients (17% and 19% of the total population, respectively), it was not possible to identify specific risk factors for the development of thromboembolic events during influenza. However, our study was not designed for such purposes.

VTE incidence varies by gender across the lifespan. In younger adulthood, women have a slightly higher annual incidence of VTE than men, but after midlife, men have a faster increase in VTE incidence than women [52]. According to the literature, twenty of our patients over the age of 40 were male, while seven were female. Surprisingly, most of the patients in the study were under the age of 50 (35/58, 60.3%), despite old age being one of the most common risk factors for venous thromboembolism [53,54,55].

Many studies in the literature indicate that Black Americans are at a higher risk of VTE than are white Americans, and Asian populations have, by far, the lowest incidence of VTE. The reasons for these potential racial differences remain unknown, with risk factors such as obesity, diabetes, and elevated factor VIII being more common in Black people, and genetic polymorphisms such as factor V Leiden and the prothrombin gene 20210A mutation being more common in white people. However, data on the incidence of and risk factors for VTE outside of Europe and North America, as well as in populations with non-European ancestry, are limited, and some of these differences may be due to a lack of surveillance for VTE, a lack of suspicion in “low risk populations”, or a lack of access to medical care, as well as different regional scenarios [56]. Unfortunately, the ethnicity of the patients was not indicated in many studies, so we cannot draw conclusions about this condition.

The most common onset symptom in our series was respiratory insufficiency (18/58 of the patients, 31.3%). Most patients had an H1N1 infection, which could indicate that this strain has a greater ability to induce thromboembolic phenomena.

Except for one case of complete superior mesenteric vein thrombosis, no pulmonary embolic events or deep-vein thrombosis were reported in the pediatric population. Stroke was the only other complication reported in this last population. In adults, on the other hand, venous thromboembolic events were the most frequently reported. This could underlie different mechanisms and risk factors for thromboembolic events in adults and children. In the current study, 36.2% of the cases, including our patient, had “de novo” pulmonary thromboembolism with no evidence of underlying DVT. This finding contrasts with the incidence of PE “de novo” reported in the literature among non-influenza patients, which ranged from 0% to 22.6% [57,58]. This suggests that local inflammation of the lung parenchyma may theoretically contribute to endothelial activation and subsequent induction of a hypercoagulable state in the surrounding microenvironment, similar to the endothelial response preceding the development of DVT in the extremities, with the potential to progress to local “de novo” thrombosis of the pulmonary vessels. This may be related to “immunothrombosis”. Moreover, during influenza virus infection, the extrinsic coagulation pathway was stimulated with a reduced generation of key inhibitors of coagulation and fibrinolysis, namely activated protein C and plasminogen—activator inhibitor type-1 [59]. However, influenza was not identified as an independent risk factor in a nested case–control study of patients suspected of having pulmonary embolism (adjusted OR 0.22, 95% CI 0.03–1.72) [1,8,59].

Data on thromboembolic risk in pregnancy are conflicting, but pregnancy and the post-partum period are known to be hypercoagulable states. In developed countries, pulmonary embolism is an important cause of maternal mortality [60,61,62]. Acute myocardial infarction is uncommon in pregnant women, especially those under the age of 30 [63]. Only three patients in our series were pregnant, two of them under 30 years old, and they had a deep venous thrombosis, an extensive cerebrovascular accident, and an acute myocardial infarction.

Screening for congenital hypercoagulable states was not performed or reported in most of the cases included in our case, so we cannot rule out the possibility that some events were caused by factors other than influenza virus infection. SARS-CoV-2 and influenza viruses share many similarities, including transmission routes and clinical presentations. The influenza virus, like SARS-CoV-2, has extensive effects on inflammatory and coagulation pathways and may be a cause of cardiovascular disease [59,64,65,66]. A significant component of susceptibility is attributed to host genetics, such as host frailty, in addition to the underlying hyperinflammatory syndrome, which is the most implicated in cardiovascular complications in both viruses [59,67,68]. Adult influenza is typically a self-limiting disease; thus, it is frequently not reported to public health authorities, resulting in underreported data [66]. Regardless, various epidemiological studies have suggested a link between cardiovascular mortality and influenza, but unlike for SARS-CoV-2 infections, there have been no studies evaluating the progression and prognostic value of cardiac markers for influenza. Although neutrophil-driven immunothrombosis is a key component of severe COVID-19, innate immunity and neutrophil extracellular trap formation (NETosis) have also been linked to influenza. Immunothrombotic vessel occlusion and NETosis, on the other hand, have been found to be significantly higher in severe SARS-CoV-2 pneumonia compared to influenza pneumonia [69]. Moreover, it should be noted that over 48% of patients had severe disease requiring invasive and non-invasive mechanical ventilation, and this could be linked to the development of thrombosis.

In conclusion, our findings are important in clarifying that thrombosis is a complication seen with influenza in adults and children. Thrombotic events can occur despite thromboprophylaxis in hospitalized patients. Symptoms of pulmonary embolism and influenza can be very similar, so a careful clinical evaluation is required for proper patient management, possible instrumental deepening with pulmonary CT angiography, and appropriate pharmacological interventions, especially for patients with respiratory failure. To better understand this phenomenon, it could be interesting to perform a molecular swab to search for influenza viruses in all cases of thromboembolic events occurring in the winter period. Furthermore, because cerebral ischemic events, myocardial infarction, and deep venous thrombosis are time-dependent pathologies, it is critical to maintain high clinical vigilance in influenza infections and promptly request brain CT scans, electrocardiograms, or arterial or venous Doppler echo-color examinations to rule them out or proceed with early treatment. This is particularly important in patients with complicated influenza in the absence of rapid clinical resolution or in cases of worsening respiratory function or the appearance of symptoms suggestive of organ damage, seemingly unrelated to influenza.

## Figures and Tables

**Figure 1 viruses-14-02817-f001:**
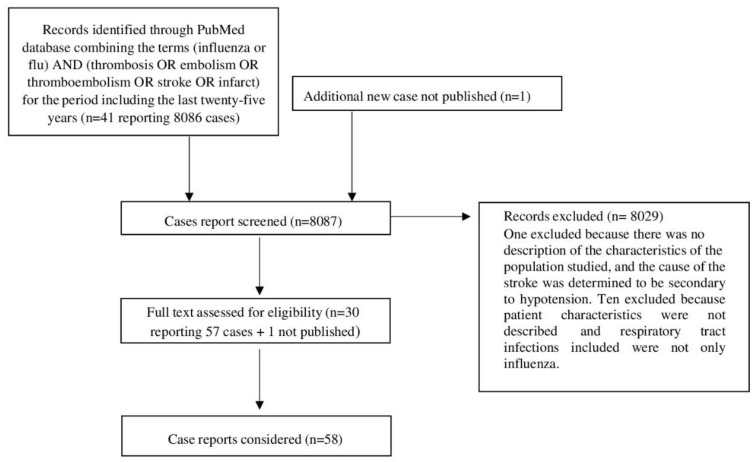
Flow chart summarizing the literature research approach.

**Table 1 viruses-14-02817-t001:** Epidemiological and clinical characteristics, complications, and outcomes of study population.

Author Nation	Sex	Age Yrs	Site of Thrombosis	Risk Factors/Comorbidities	Other Complications	Outcome
Ulvestad E., 2000	F	22	Pulmonary embolism	Siögren syndrome, anti-phospholipid syndrome	None	Survived
Ohrui T., 2000	F	84	Bilateral microembolism pulmonary arteries	Mild well-controlled hypertension	None	Survived
Ohrui T., 2000	M	81	Bilateral microembolism pulmonary arteries	Benign prostate hypertrophy	None	Survived
Bell M.L., 2004	M	4	Focal partial occlusion of left middle cerebral artery origin	None		Survived with sequelae (speech disturbance and left occipital epileptiform activity)
CDC, 2009	M	28	Pulmonary embolism	Asthma, obesity	ARDS †	Death
CDC, 2009	M	21	Pulmonary embolism	Obesity	ARDS †	Survived
CDC, 2009	M	35	Pulmonary embolism	Obesity	ARDS †	Survived
CDC, 2009	M	43	Pulmonary embolism	Obesity	ARDS †	Death
CDC, 2009	M	52	Pulmonary embolism	None	ARDS †	Survived
Harms P.W., 2010	M	28	Pulmonary embolism (antemortem diagnosis)	Asthma, hypertension, obesity	ARDS †	Death
Harms P.W., 2010	M	43	Pulmonary embolism (antemortem diagnosis)	Hypertension, obesity	ARDS †, diarrhea caused by *Clostridioides difficile*	Death
Harms P.W., 2010	M	57	Pulmonary embolism (right lung) postmortem diagnosis	Non-Hodgkin lymphoma in remission, chronic renal failure	ARDS, bacterial pneumonia (*Acinetobacter baumanii*)	Death
Harms P.W., 2010	M	51	Deep venous thrombosis (femoral/iliac) (antemortem diagnosis)	Sarcoidosis, chronic use of prednisone, obesity	ARDS †	Death
Harms P.W., 2010	M	23	Pulmonary embolism postmortem diagnosis	Autism, seizures	ARDS †, MSSA ‡ (blood)	Death
Harms P.W., 2010	M	31	Portal vein thrombosis (antemortem diagnosis)	OSA, COPD, Schizophrenia, hypertension, obesity, smoker	ARDS †, MRSA § (blood, nose, tracheostomy), VRE ¶ (tracheostomy)	Death
Harms P.W., 2010	M	57	Pulmonary embolism (right lower lobe) (postmortem diagnosis)	Obesity	ARDS †, Enterococcus faecium (lung culture)	Death
Bunce P.E., 2011	F	50	Infrarenal aorta thrombosis	Not reported	ECMO ††	Survived with sequelae (left-sided above knee amputation)
Bunce P.E., 2011	M	49	ST-elevation myocardial infarction	Not reported	None	Death
Bunce P.E., 2011	M	35	Bilateral DVT (femoral and iliac), presumed pulmonary embolism	Not reported	None	Death
Bunce P.E., 2011	F	18	Left arm DVT	Not reported	None	Survived
Bunce P.E., 2011	M	46	Right external iliac vein and common femoral vein thrombosis	Not reported	None	Survived
Bunce P.E., 2011	M	84	Pulmonary embolism	Not reported	None	Survived
Bunce P.E., 2011	M	39	ST-elevation myocardial infarction	Not reported	None	Survived with sequelae (developed left ventricle thrombus)
Mansurali N., 2011	F	not reported	Pelvic deep-vein thrombosis	None	Non-traumatic subperiosteal hemorrhage, critical care polyneuropathy and dialysis-dependent acute renal insufficiency, ECMO ††	Survived
Frobert E., 2011	F	6.5	Basilar artery with thalami ischemic infarction	None	Bacterial pneumonia (*Mycoplasma pneumoniae*)	Survived with sequelae. Locked-in syndrome (spastic tetraplegia, axial hypotonia, areactivity, few eye blinks)
Hayakawa T, 2011	F	5	Superior mesenteric vein complete thrombosis	Type 2 protein S deficiency	None	Survived, resection of ischemic small intestine
Krummel-McCracken K., 2011	F	34	Multiple acute and subacute strokes throughout both hemispheres	Mental retardation, hypothyroidism, hypertension	ARDS †	Survived with sequelae (gait disturbance)
Citton R., 2012	M	42	Pulmonary artery embolism, a right lower limb deep phlebothrombosis	None	Bacterial pneumonia (*Legionella*), subphrenic abscess	Survived, splenectomy for splenic rupture caused by diffuse intravascular coagulation
Burad J., 2012	F	50	Multiple brain infarcts (right middle cerebral artery territory and cerebellum)	Mild schizophrenia	ARDS †	Survived with sequelae (left-sided weakness)
Gökçe M., 2012	F	16	Left leg deep venous thrombosis	Cerebral palsy due to periventricular leukomalacia as complication of premature birth, surgery for developmental dysplasia of the left hip without thrombotic events	None	Survived
Honorat R., 2012	F	<1	Acute infarct in the left middle cerebral artery territory	None	None	Survived, discrete cortical residual atrophy
Simon M., 2013	M	26	Cerebral (superior sagittal sinus) thrombosis	None	Encephalitis	Death
Calzedda R., 2014	F	2	Massive ischemic area in the right parieto-temporal region and in the cerebral peduncle	None	None	Survived with sequelae (left arm movement impairment and a slight limp)
Iwanaga N., 2014	M	52	Stenosis of the left anterior descending coronary artery and myocardial infarction	Chronic ischaemic heart disease, diabetes, smoker	Myocarditis	Death
Cheung A.Y., 2015	F	13	Bilateral ischemic maculopathy (angiography)	Low protein S activity	Encephalitis	Survived with sequelae (visual acuity remained at light perception bilaterally)
Avnon L.S., 2015	F	19	Bilateral cerebrovascular event	Pregnancy, 21 weeks	Bacterial pneumonia (Streptococcus pneumoniae), spontaneous abortion, polyneuropathy, bilateral barotraumas	Survived with sequelae (left hemiparesis and aphasia)
Avnon L.S., 2015	F	28	Acute anterior ST elevation	Pregnancy, 24 weeks	None	Death
Avnon L.S., 2015	F	40	Subclavian and jugular DVT	Pregnancy, 36 weeks, emergency caesarean section	Barotraumas	Death
Avnon L.S., 2015	M	23	Left subclavian DVT	B-cell lymphoma (recent induction chemotherapy), cerebral palsy	Neutropenic fever after chemotherapy, bilateral barotraumas, *Staphylococcus aureus* furunculosis of the skin, Guillan Barrè	Survived
Avnon L.S., 2015	M	22	Right leg DVT	Acute myeloid leukemia	None	Survived
Breker D.A., 2015	F	13	Ischemic lesions of both lateral geniculate nuclei and in the cerebellar vermis and dorsal midbrain, plus retinal ischemia at ophthalmoscopy in both eyes	None	None	Survived with sequelae (vision impairment)
Dimakakos E., 2016	F	49	Pulmonary embolism	Dyslipidemia, hypothyroidism	None	Survived
Dimakakos E., 2016	M	38	Pulmonary embolism	Smoker	None	Survived
Dimakakos E., 2016	M	47	Pulmonary embolism + DVT	In hospital immobilization	None	Survived
Dimakakos E., 2016	M	73	Pulmonary embolism	Coronary artery disease, smoker	None	Survived
Dimakakos E., 2016	M	54	Pulmonary embolism	Dyslipidemia	None	Survived
Dimakakos E., 2016	M	47	Pulmonary embolism	Smoker	None	Survived
Dimakakos E., 2016	M	49	Pulmonary embolism	Hypertension, smoker	None	Survived
Huzmeli C., 2016	M	28	Infrarenal aorta thrombosis, bilateral iliac artery thrombosis	Heart failure, pacemaker carrier, protein S activity low	Chronic renal failure, *Acinetobacter baumanii* sepsis	Death
Collins M.H., 2016	M	44	Superior mesenteric vein thrombosis, which extended into the main portal vein and proximal splenic vein	None	Intrahepatic hematoma requiring subselective hepatic artery embolization	Survived
Cho S.H., 2016	F	32	Acute ischemic stroke (bilateral thalamus, midbrain, pons, cerebellum)	None	None	Not reported
Taniguchi D., 2017	F	52	Deep cerebral venous thrombosis-occlusion of internal cerebral vein, the great vein of galen and the straight sinus-(RMN and angiography), pulmonary embolism	None	None	Survived with sequelae (slight memory and mild right-sided hemiparesis)
Arbit B., 2017	M	21	Total occlusion of the left anterior descending artery and acute myocardial infarction	Cocaine abuse five weeks prior to the onset of the flu-like symptoms, smoker	None	Survived
Cascio A., 2018 (our case)	M	38	Pulmonary embolism	None	None	Survived
Ishiguro T., 2019	M	58	Deep venous thrombosis, acute pulmonary artery embolism, acute arterial embolism, left ventricular thrombus	Ischemic heart disease, hypertension, smoker	None	Survived with sequelae (amputation of the right foot)
Javedani P.P., 2019	F	<1	Right cerebral hemisphere infarction and cortical veins thrombosis	Intrauterine drug exposure	Seizures	Survived (discharged on levetiracetam)
Oh GM, 2020	M	64	Superior mesenteric vein thrombosis	Dyslipidemia, hypertension	None	Survived
Mizutani K., 2020	F	40	Simultaneous cerebral multifocal infarctions	CADASIL syndrome	None	Survived

† Acute respiratory distress syndrome; ‡ Methicillin susceptible *Staphylococcus aureus*; § Methicillin resistant *Staphylococcus aureus*; ¶ Vancomycin resistant enterococci; †† Extracorporeal membrane oxygenation.

**Table 2 viruses-14-02817-t002:** Comorbidities and sites of thromboembolic events in the study population.

Site of Thromboembolic Event									
		DVT * 12/58 (20.6%)	PE ** “De Novo” 21/58 (36.2%)	DVT + PE 3/58 (5.1%)	Stroke 11/58 (18.9%)	Acute Myocardial Infarction 5/58 (8.6%)	Arterial Thrombosis, Excluding MI and Stroke 4/58 (6.8%)	DVT + PE + Arterial Thrombosis + Acute Myocardial Infarction 1/58 (1.7%)	PE + Cerebral Thrombosis 1/58 (1.7%)
SEX	n° (%)								
Male	36 (62)	7	18	3	2	4	1	1	0
Female	22 (37.9)	5	3	0	9	1	3	0	1

COMORBIDITIES	n°								
Hypertension	4	1	2	-	1	-	-	-	-
Obesity	6	1	5	-	-	-	-	-	-
Diabetes	3	1	1	-	-	1	-	-	-
Pregnancy	3	1	-	-	1	1	-	-	-
Neoplasia	3	2	1	-	-	-	-	-	-
Coagulation disorders	3	1	-	-	-	-	2	-	-
Smokers	4	-	2	-	-	2	-	-	-
Obesity + hypertension	3	-	3	-	-	-	-	-	-
Hypertension + smoking habit	3	-	2	-	-	-	-	1	-
Hypertension + obesity + smoking habit	1	-	1	-	-	-	-	-	-
Autoimmune diseases	1	-	1	-	-	-	-	-	-
Immunosuppression	5	4	1	-	-	-	-	-	-
Ischemic heart disease	3	-	1	-	-	1	-	1	-
Dyslipidemia	3	1	2	-	-	-	-	-	-
Chronic kidney disease	1	-	1	-	-	-	-	-	-
Asthma/COPD ***	3	1	2	-	-	-	-	-	-
No comorbidities/not reported	25	4	5	3	8	2	2	-	1
Drug users	2	-	-	-	1	1	-	-	-
Outcome (dead)		3 (25%)	7 (33.3%)	1 (33.3%)	1 (9%)	3 (60%)	1 (20%)	0	0

* DVT—deep venous thrombosis ** PE—pulmonary embolism. *** Chronic obstructive pulmonary disease.

## Data Availability

All data used and/or analyzed during this study are included in this published article.

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
