# Peer review of "Thromboembolic Events in Patients with Influenza: A Scoping Review"

_viruses, 2022, doi:10.3390/v14122817_

Round 1

Reviewer 1 Report

Dear Authors

Well done on collecting this interesting literature review on thrombosis in influenza. I make the following suggestions for your manuscript:

Abstract L20 - Suggest changing this to "58 cases with..... were reviewed." so your case is included

Case Report L71-84 - For the purpose of this journal, do abbreviations/acronyms need to be expanded with their use. Change to pH on line 79 and change to 'was not' on L85. Weight should also be included as relevant for heparin dosing.

Case Report L96 - Were sequential D-Dimers done for persisting hypoxia despite antiviral treatment? This should be made clear.

Case Report - I think it needs to be highlighted that this patient has PE in the context of a severe influenza-related pneumonia on thromboprophylaxis, which prompted the further literature review. This could either be at the end of this section or at the beginning of the Discussion

Table 1 - There are multiple inconsistencies e.g pulmonary (embolism), the inclusion of thrombus in some descriptions and not in others. There are also non-English grammatical/spelling errors e.g low Protein S level, bilateral cerebrovascular events, bacterial pneumonia, subclavian and jugular thrombosis, barotrauma, left subclavian DVT, left ventricular thrombus, hemisphere. Also the abbreviation LGN should be expanded.

Results - There is inconsistent use of % and percent. I would suggest changing all to %

Results L172-173 - Change to 'and had a pacemaker'

Results L198 - What does 'VN' mean? Is this meant to be normal values?

Results - It would be useful to know what percentage of patients were admitted to intensive care as this give a greater impression if illness severity was related to thrombosis development in influenza.

Table 2 - Should the column for arterial thrombosis state '(excluding MI and stroke)'?

Discussion L261-271 - I am unclear what this paragraph adds to the article as you have not discussed ethnicity in the rest of the article.

Discussion L286 - I would suggest adding that this may be related to 'immunothrombosis'. I appreciate that the authors have discussed this in comparison to COVID-19 later in the discussion

Discussion L273 - Remove 'A'?

Discussion L288 - Change to 'reduced generation of key inhibitors of coagulation and fibrinolysis, activated protein C and plasminogen-activator inhibitor type-1'. Also, this needs to be referenced

Discussion L317 - I am not sure if you can draw this conclusion. Your literature review has demonstrated a variety of thrombotic complications with influenza but you have not demonstrated incidence. In fact, this conflicts with your comments on L289. I would suggest it is rephrased to suggest that thrombosis is a complication seen in influenza. I would also suggest highlighted that thrombotic events can occur despite thromboprophylaxis in hospitalised patients.

Reviewer 2 Report

A very well done and written scoping review of the topic. I have one major concern that needs to be addressed by the authors. To my mind the article combines three different patient populations with probable different p mechanisms and risk factors for thromboembolic events. Children and arterial and venous thrombosis in adults. This is manifested by the fact that there were no PE or DVTs in the pediatric population and in the adults VTE were commonly reported. 

Some other issues

1. I am not sure what the detailed case report added to the paper

2. There was no mention of the quality of the Journals from which the cases were taken

3. I would like the authors to expand on what is the take home message of the paper and I am not sure I understand what they mean by instrumental deepening in their conclusion. 

4. The authors also do discuss the mechanism for VTE in influenza and does the disease act differently in the influenza patient. 

Round 2

Reviewer 2 Report

revisions accepted